# Untangling the Conformational Plasticity of V66M Human proBDNF Polymorphism as a Modifier of Psychiatric Disorder Susceptibility

**DOI:** 10.3390/ijms23126596

**Published:** 2022-06-13

**Authors:** Sonia Covaceuszach, Leticia Yamila Peche, Petr Valeryevich Konarev, Joze Grdadolnik, Antonino Cattaneo, Doriano Lamba

**Affiliations:** 1Istituto di Cristallografia, Consiglio Nazionale delle Ricerche, 34149 Trieste, Italy; lypeche@gmail.com; 2A.V. Shubnikov Institute of Crystallography of Federal Scientific Research Centre “Crystallography and Photonics” of Russian Academy of Sciences, 119333 Moscow, Russia; peter_konarev@mail.ru; 3Laboratory for Molecular Structural Dynamics, Theory Department, National Institute of Chemistry, Hajdrihova 19, 1001 Ljubljana, Slovenia; joze.grdadolnik@ki.si; 4European Brain Research Institute, 00161 Roma, Italy; antonino.cattaneo@sns.it; 5Scuola Normale Superiore, 56126 Pisa, Italy; 6Consorzio Interuniversitario “Istituto Nazionale Biostrutture e Biosistemi”, 00136 Roma, Italy

**Keywords:** proBDNF, V66M polymorphism, structural characterization, conformational plasticity, neuropsychiatric disorders

## Abstract

The human genetic variant BDNF (V66M) represents the first example of neurotrophin family member that has been linked to psychiatric disorders. In order to elucidate structural differences that account for the effects in cognitive function, this hproBDNF polymorph was expressed, refolded, purified, and compared directly to the WT variant for the first time for differences in their 3D structures by DSF, limited proteolysis, FT-IR, and SAXS measurements in solution. Our complementary studies revealed a deep impact of V66M polymorphism on hproBDNF conformations in solution. Although the mean conformation in solution appears to be more compact in the V66M variant, overall, we demonstrated a large increase in flexibility in solution upon V66M mutation. Thus, considering that plasticity in IDR is crucial for protein function, the observed alterations may be related to the functional alterations in hproBDNF binding to its receptors p75NTR, sortilin, HAP1, and SorCS2. These effects can provoke altered intracellular neuronal trafficking and/or affect proBDNF physiological functions, leading to many brain-associated diseases and conditions such as cognitive impairment and anxiety. The structural alterations highlighted in the present study may pave the way to the development of drug discovery strategies to provide greater therapeutic responses and of novel pharmacologic strategy in human populations with this common polymorphism, ultimately guiding personalized medicine for neuropsychiatric disorders.

## 1. Introduction

Among the potential genetic-risk factors, non-conservative polymorphisms in the *BDNF* gene, which alter the distribution and function of brain-derived neurotrophic factor (BDNF), have been strongly associated to neuropsychiatric disorders [1,2,3].

BDNF is a member of the neurotrophin superfamily involved in the regulation of proliferation, maturation, and maintenance of the neuronal function in specific brain areas [4]. The interaction with its cognate tropomyosin kinase receptor B (TrkB) and/or p75 pan-neurotrophin receptor (p75NTR) [5] leads to a series of intracellular signalling cascades mediating neuronal survival, axonal and dendritic growth and guidance, synaptic structure and connections, neurotransmitter release, capture and synaptic tagging (such as long-term potentiation), learning and memory, and emotional behaviour [4,6].

Widely expressed during the development and in adult mammalian central nervous system, BDNF has been associated to the pathophysiology of affective disorders and to the antidepressants and mood stabilizers psychopharmacology [7,8,9].

Encoded by *BDNF* gene on the short arm of chromosome 11 (11p14.1), BDNF is synthesized as a precursor homo-dimeric protein (preproBDNF) and released both as proBDNF and mature BDNF through constitutive secretion or by activity-dependent exocytosis in excitable cells secreting granules [10]. ProBDNF is proteolytically cleaved into BDNF by intracellular furin/proprotein convertases and extracellular proteases (plasmin/matrix metallopeptidases) [11].

proBDNF is not just an inactive precursor where the propeptide domain acts as an intramolecular chaperone and facilitates folding under oxidative conditions but serves as a signalling protein exerting opposing effect on synaptic structure, synaptic plasticity, synaptic transmission, and behaviour. The binding of BDNF to TrkB activates different signalling cascades and leads to neuron survival and plasticity, whereas the interaction of proBDNF to p75NTR/sortilin receptor complex has been highly involved in apoptosis [12].

Recently, a frequent single nucleotide polymorphism in the human *BDNF* gene (dbSNP number rs6265) has been identified at nucleotide 196 (G/A), leading to a non-conservative mutation at codon 66 in the pro-domain (V66M). The frequency of this polymorphism varies depending upon ethnicity [13] and has been shown to lead, in healthy young carriers, to a volumetric reduction in specific brain areas including the hippocampus and thus to hippocampal-dependent memory deficits [14,15,16,17].

Earlier studies demonstrated that a single M66 allele acts in a dominant way by impairing trafficking to the distal dendrites and thus leading to a reduction in distribution into neuronal dendrites. In addition, it decreases targeting to secretory granules and leads to subsequent impairment in activity-dependent release of mature BDNF in neurons [14,18,19] and defects in dendritic targeting of BDNF mRNA [14,18,20,21]. These impairments are due to the formation of heterodimers that are less efficiently sorted into the regulated secretory pathway [18] and this finding suggests the existence of a specific trafficking signal in the pro-domain region encompassing the codon 66. It is still unclear whether the molecular basis of these impairments is ascribed to impaired binding to the receptors p75NTR [22,23], sortilin [22], Huntingtin associated protein-1 (HAP1) [24], sortilin-related Vps10p-domain sorting receptor 2 (SorCS2) [25], and/or to altered intracellular neuronal signalling.

Interestingly, there is growing evidence that this polymorphism is associated to the pathophysiology of neuropsychiatric and neurodegenerative disorders that makes individuals more susceptible to bipolar disorders as well as to Alzheimer’s disease, Parkinson’s disease, depression, obsessive-compulsive disorders, eating disorders, schizophrenia, anxiety, addiction, and memory and learning disabilities [26,27,28,29,30,31,32,33,34,35].

There are only very sparse reports suggesting that the V66M polymorphism induces structural changes of proBDNF. To this regard, a detailed structural characterization is of fundamental importance to unveil the mechanisms that are involved in physiological protein functions [36] and the molecular determinants of the pathological variants that disrupt these mechanisms [37].

In silico prediction unveiled the likely occurrence of naturally disordered regions in either proBDNF, and its V66M variant [38,39]. Extended molecular dynamics simulations on a 3D ab initio generate model and homology modelling prompted the V66M polymorphism [40] to affect the essential motions, the hydrogen-bonding network, and the local and non-local secondary structure conformation of the proBDNF pro-domain. Lastly, a structural bioinformatics and molecular dynamics simulation study [41] pointed out the V66M mutation to induce instability in hproBDNF-HAP1 and hproBDNF-Sortilin1 complexes.

Anastasia et al. [25] and Kailainathan et al. [38] exploited NMR and Far UV CD spectroscopies, respectively, to experimentally assess the structural differences occurring between the prodomains of hproBDNF WT and V66M polymorphs. These studies confirmed that either proteins lack ordered secondary or tertiary structures.

SAXS-driven ensemble modelling has previously been successfully employed to gain insights into the conformational plasticity of mouse proNGF [42,43] and of the full-length human proneurotrophins family members [44].

The present comparative study aims, for the first time, to resolve the biochemical and biophysical properties of hproBDNF and of the V66M variant in solution by exploiting integrative biophysical and structural biology tools, including limited proteolysis, DSF, FT-IR, and SAXS.

## 2. Results and Discussion

We employed a combined biochemical-biophysical approach in order to evaluate the impact of V66M polymorphism on hproBDNF conformational plasticity in solution.

We expressed recombinant WT and V66M hproBDNF variants in bacterial inclusion bodies, pulsed-refolded and purified according to Covaceuszach et al. [44]. The absence of glycosylation in this bacterial expression system ensures the full homogeneity of the expressed proteins, an important property for biophysical and structural studies. Notably, it has been demonstrated that there are no functional differences between glycosylated proBDNF samples produced by baculovirus system and its non-glycosylated form produced by *E. coli* [45].

### 2.1. V66M Polymorphism Influences the Conformation of hproBDNF in Solution

A previous far-UV CD comparative study [38] showed that both WT hproBDNF and the V66M variant spectra had similar shapes in the far-UV region with a minimum peak at ~204 nm, indicative of a predominant β-sheet structure. Spectral overlay further showed proBDNF V66M to contain a small increase in negative ellipticity at ~222 nm, the region characteristic for helical proteins.

Our present Fourier transform infrared spectroscopy (FT-IR) experiments and a thorough analysis of the amide I (1600–1700 cm^−1^) and amide II (1510–1580 cm^−1^) regions (Figure 1A,B, for hproBDNF WT and the V66M variant, respectively) do not reveal the presence of helical content in either of the proteins [46,47]. The band decomposition of amide I reveals the bands at 1680 cm^−1^ and 1633 cm^−1^ characteristic of β-sheet structures. However, α-helical and disordered structures absorb almost at the same wavenumber. The analysis of the amide III region (1200–1350 cm^−1^) shown in Figure 1C clearly indicates that the band between 1650 and 1660 cm^−1^ is due to the presence of disordered rather than helical structure. A variation integral peak intensity has been detected in the bands at 1680 cm^−1^ and 1633 cm^−1^, ascribable to the β sheet conformation (54% and 57% for the WT and V66M variant, respectively) and in the bands near 1655 cm^−1^ assigned to the unordered part of the protein (38% and 39% for the WT and V66M variants, respectively). All these small but significant differences point to a structural rearrangement induced by the V66M mutation in the hproBDNF overall conformation.

Indeed, limited proteolysis experiments (Figure 1D) highlight that a difference occurs in the accessibility for the thermolysin, actinase, and trypsin proteases being used. A faster and complete degradation of the WT protein with respect to the V66M variant strongly supports the hypothesis that the V66M mutation induces a conformational change compatible with a more compact overall shape that would mask most of the accessible proteolytic cleavage sites.

SAXS provides unique insights into conformational changes in proteins and in deciphering protein flexibility, of paramount importance in the structural characterization of intrinsically disordered proteins.

In Table 1, the overall parameters derived from the SAXS data are summarized, compared to those previously reported for hproBDNF WT [44].

The linearity of the Guinier plots at very small angles, i.e., *s* < 1.3/R_g_, (see the insert in Figure 1E) confirmed the homogeneity of the sample, despite the occurrence of a rather small degree of aggregation. The derived experimental scattering patterns of V66M did not show any concentration-dependent self-association. The processed scattering data is shown in Figure 1E, and the corresponding computed distance distribution function p(r) is compared to the one derived for the WT protein [44] (Figure 1F).

The excluded Porod volume V_p_ of 85,675 Å^3^ and derived MM value of 50.4 kDa well agree with values expected for homo-dimeric species, being, according to empirical findings [48], the V_p_ for globular proteins 1.5–1.7 times larger than their MM.

The hproBDNF V66M polymorph is characterized by a significant shrinkage of the radius of gyration (R_g_) (Table 1) (35.50 ± 0.05 Å vs. 37.20 ± 0.06 Å for the WT protein [44]). Likewise, the profile of the computed distance distribution functions p(r), characterized by a single peak with a tail typical of proteins with elongated structures, point to a size decrease in the maximum dimensions of the missense mutant (115 ± 5 Å for V66M vs. 125 ± 6 Å for the WT protein [44]).

Overall, these data suggest that V66M polymorphism induces a genuine conformational change of hproBDNF towards an overall more compact shape.

The CORAL approach [48] has been used for the low-resolution modelling of the un-structured loops and termini of the WT hproBDNF and V66M variant pro-peptide domains in the absence of high-resolution structures. Multiple runs were performed, yielding variable conformations, all providing good fits to the experimental data (i.e., χ^2^ 1.08 ± 0.03). The variety of configurations points to significant flexibility of the pro-peptide region (see below) with mean values of NSD of 2.49 ± 0.92. Nevertheless, the good fit of the best rigid-body model (respectively χ^2^ 1.09 with NSD = 1.76) suggests that the models shown in Figure 2 can be considered as reliable representations of the respective average conformations.

These results were compared with the ab initio model obtained for the WT protein [44] and proves the V66M polymorphism to affect the structural flexibility in solution of the hproBDNF pro-domain leading to an overall more compact averaged structure.

### 2.2. V66M Polymorphism Affects the Thermostability and the Conformational Plasticity of hproBDNF in Solution

Differential scanning fluorimetry (DSF) allows us to monitor protein thermal denaturation by using an environmentally sensitive high quantum yield fluorescent dye (SYPRO Orange). The subsequent analysis of the first derivative of the melting curve allows us to calculate T_m_, a parameter that quantifies thermal and conformational stability. Thermo-melting profile of the V66M mutant protein was compared to the one measured for the WT protein [44] (Figure 3A).

Both curves are being characterized by a very low and flat fluorescence background in the pre-transition region, and a bimodal profile, typical of proteins with more than one domain. In detail, the second sharp transition at high T (T_m_ 75.3 ± 0.5 °C and 74.9 ± 0.7 °C for WT hproBDNF and V66M variant, respectively) is unchanged in the two melting curves and is probably due to the unfolding of the globular domain of the mature hBDNF. On the contrary, the transition at low T shows a different behavior. The profile of V66M polymorph results to be sharper and therefore more cooperative with a significant decrease in the T_m_ (from 35.3 ± 0.3 °C compared to 38.3 ± 0.4 °C reported for the WT protein [44]).

It is tempting to hypothesize that the structural plasticity of the hproBDNF V66M variant, inferred by physicochemical factors such as the re-arrangement of the hydrogen bonding networking, as well as hydrophobic and/or salt bridges intra-molecular interactions, might in turn confer lower thermostability. Overall, these may be especially critical in vivo, being the T_m_ of hproBDNF V66M variant well below the human body temperature.

SAXS is not only a structural biology tool that probes the size and shape of disordered proteins, but it is also a powerful tool able to provide insights on the inherent plasticity of proteins, being the latter a pre-requisite to exert their prescribed biological functions. This is particularly crucial in the case of intrinsically unstructured proteins, where a comprehensive understanding of protein function entails a characterization of protein flexibility both in physiological and pathological contexts.

In detail, Kratky plot is an extremely useful representation of the scattering intensity for qualitative assessment of the protein disorder without any modelling; *I*(*s*) decays as *s*^−4^ for compact proteins, whereas the scattering intensity of a flexible Gaussian chain decays as *s*^−2^ or slower [49]. Thus, while compact proteins have *s*^2^*I*(*s*) values that approach zero (or baseline) at high *s*, unfolded or disordered proteins, such as the two variants of hproBDNF, will generally have a plateau at intermediate angles followed by continuously increasing values of *s*^2^*I*(*s*) at wide angles [50,51,52]. This behavior seems to be even more pronounced in the V66M variant (Figure 3B).

In order to perform a quantitative analysis of the impact of V66M polymorphism on flexibility and size distribution of possible multiple configurations in solution of hproBDNF, the ensemble optimization method (EOM) [53] has been exploited. It allows deriving accurate ensemble models of flexible proteins, which collectively describe the experimental profile with a good agreement with respect to the experimental scattering curve (i.e., χ^2^ of 0.967).

The availability of such ensemble methods has greatly advanced the study of flexible proteins by SAXS. Indeed, they provide a description in terms of the statistical distributions of structural parameters or conformations, greatly improving the traditional analyses based on averaged parameters extracted from raw data that simply describe the mean behavior of the macromolecule in solution. Most importantly, ensemble methods such as the ensemble optimization method (EOM) allow monitoring structural perturbations exerted by mutations [54] in terms of ensembles of conformations.

Figure 3D compares the results of the EOM analysis for the V66M variant to the one previously reported for the WT protein (Figure 3C) [44], in terms of size distribution, by plotting the R_g_ of the structures representing the initial random pool and the selected ensembles.

In both cases, the R_g_ distribution of the selected ensemble (solid lines) is wider than the distribution of the randomly generated models, a typical hallmark of intrinsically disordered proteins. Nevertheless, the two distributions show a major difference in their respective profiles.

The V66M polymorphism greatly affects the distribution (Figure 3D solid grey line). Although still roughly bimodal, it spans over a range of dimensions much wider than the distribution of the WT variant. Indeed, the main peak is broader and asymmetric, instead of being sharp and symmetric as for the WT protein (Figure 3C solid black line) and contains more compact conformations with respect to the WT protein. Indeed, there is a shift in the range of R_g_ values from 22–42 Å (accounting for 62% of the total population for the WT protein) to 20–50 Å (accounting for 60.5% of the total population for the V66M variant). Furthermore, the second peak of the distribution is also being affected by the V66M polymorphism. In the WT protein the second peak is broad and asymmetric (accounting for 38% of the total population) with a tail encompassing elongated conformations (accounting for 0.5% of the total population), with R_g_ values spanning between 42 and 57 Å and having a maximum around 49 Å. Instead, in the V66M polymorphism the peak is broad but symmetric, with R_g_ values in the range between 50 and 72 Å (accounting for 39.5% of the total population) and a maximum around 60 Å. The peak is being shifted to more extended models with respect to the WT protein profile. It accounts for very elongated conformations that are much less abundant in the case of the WT protein.

Thus, even if the overall parameters suggest an average conformation that is more compact because of the V66M polymorphism, the EOM analysis offer snapshots of the likely equilibrium conformational ensembles of hproBDNF that occur in solution.

## 3. Conclusions

To the best of our knowledge, this study presents the results of the first throughout structural and biophysical characterization of human WT hproBDNF and of the V66M variant related to several psychiatric disorders, with findings that may have crucial implications for its physiology.

In short, limited proteolysis experiments highlight a faster and complete degradation of hproBDNF with respect to the hproBDNF V66M variant; SAXS analysis clearly pinpoints to the conformational flexibility and structural plasticity of both the WT hproBDNF and the V66M variant, typical of intrinsically disordered proteins. Notably, the hproBDNF V66M variant adopts an overall more compact architecture. This finding well complements the observed enzymatic degradation profiles as well as previously reported binding affinity surface plasmon resonance measurements, supported by computational docking simulations, which highlighted V66M polymorphism stabilizing the interaction of BDNF with its pro-peptide (K_D_ of 0.42 nM and 0.12 nM of the WT and V66M pro-peptide domain, respectively) [55].

The prodomain of the hproBDNF V66M variant, despite its more compact structure, surprisingly shows to be less thermostable than the WT one. This apparent inconsistency due to the conformational flexibility, likely results in differences between physicochemical factors such as hydrogen bonding, hydrophobic, as well as salt bridges interactions.

Overall, the present study supports the hypothesis that the conformational flexibility and the structural plasticity of the V66M polymorphism represents a crucial structural determinant, which paves the way to a multifaceted understanding of TrkB receptor binding, signaling stability, processing, trafficking, and secretion (both constitutive and activity-dependent) of mature BDNF. These functional processes may in turn have a deep impact on brain functional physiology in neuropsychiatric disorders.

Although we are aware of the functional in vivo limitations of the present investigation, as on the other hand for most of the structural studies, these in vitro results strongly support and strengthen the functional differences and the associated behavioral consequences of the V66M hproBDNF polymorph.

Ongoing structural studies of a recently discovered and less characterized hproBDNF SNP, i.e., M122T, found in a small consanguineous family with intellectual disability [56], would allow us to confirm our present in vitro findings i.e., how a single amino acid mutation could lead to profound changes in structural stability and conformational flexibility. Furthermore, Förster resonance energy transfer experiments should help elucidating differences in the dynamics of intra-molecular interactions of hproBDNF WT and of its pathological variant/s in living cells.

We plan to complement our reported proteolytic sensitivity studies by exploiting physiological proteases involved in hproBDNF maturation, i.e., prohormone convertase, furin, plasmin, and MMP-7.

## 4. Materials and Methods

### 4.1. Fourier Transform Infrared Spectroscopy (FT-IR)

The FT-IR spectra were recorded on a Bruker Vertex 80 spectrometer equipped with the Golden Gate ATR. Typically, 64 spectra were recorded at room temperature with a resolution of 4 cm^−1^. hproBDNF WT and V66M mutant proteins at 131 μM and 38 μM concentrations, respectively, were measured in 50 mM Na phosphate, 1 mM EDTA, 150 mM NaCl pH 7. The subtraction of the sample and the scaled buffer spectra revealed the amide I, amide II, and amide III absorption bands from the protein. The spectra were processed using the OPUS software (BRUKER, Billerica, MA, USA) and the band fitting was done using Grams (Thermo Fisher Scientific, Waltham, MA, USA). The model bands were calculated using the sum of Lorentzian and Gaussian functions and optimized for height, frequency, half-width, and band shape.

### 4.2. Limited Proteolysis

Limited proteolysis of hproBDNF WT and V66M mutant proteins was carried out at 37 °C at a final concentration of 25 μM in 50 mM Sodium Phosphate pH 7.0 by using a panel of three selected proteases from Proti-Ace kits (Hampton Research, Aliso Viejo, CA, USA), i.e., actinase E, thermolysin, and trypsin at a final concentration of 2 μg/mL. The digestions were quenched after 1, 5, 10, 20, 40, and 60 min by the addition of SDS-PAGE sample buffer to aliquots of the reaction mixtures. The samples were analyzed by performing 12% SDS-PAGE and Coomassie staining.

### 4.3. Differential Scanning Fluorimetry (DSF)

DSF experiments were performed using a CFX96 Touch Biorad Real-Time PCR system (Bio-Rad, Hercules, CA, USA), setting the λ_ex_ = 470–505 nm and the λ_em_ = 540–700 nm. The protein stock in 50 mM Sodium Phosphate buffer pH 7.0 was mixed at a final concentration of 20 μM with 90× SYPRO Orange (Sigma, St. Louis, MO, USA). Fluorescence was measured with temperature increments of 0.2 °C/min in a temperature range of 20–90 °C. Experiments were performed in triplicate. The melting temperatures (T_m_), taken as inflection points of the transition curves, were calculated using Boltzmann Sigmoid fit.

### 4.4. Small-Angle X-ray Scattering Data Collection and Processing

Similar to what was reported for the hproBDNF WT protein [44], SAXS measurements at five different concentrations (the ranges are reported in Table 1) of hproBDNF V66M mutant protein in 50 mM Sodium Phosphate pH 7.0 were performed at PETRAIII/DESY [57] on the P12 beamline EMBL SAXS-WAXS (Hamburg, Germany). Data were collected as 20 × 0.05 s exposures using a PILATUS (Dectris, Ltd., Dättwil, Aargau, Switzerland) 2 M pixel X-ray detector; sample-detector distance 3.00 m; wavelength 1.24 Å. No radiation damage was detected, by comparing the scattering profiles for the collected frames.

After normalization to the intensity of the transmitted beam, for each of the samples the collected frames were merged. Subtraction of the buffer’s contribution to the scattering and further processing steps were performed using the program PRIMUS [58] from the ATSAS data analysis suite [59]. The radius of gyration R_g_ and the forward scattering I(0) were computed by the Guinier approximation [60] assuming that at very small angles (0 < *s* < 1.3/R_g_) the intensity is represented as *I*(*s*) = I(0) exp(−1/3(R_g_·*s*)^2^). GNOM [61] was used to calculate the maximum sizes D_max_ and the pair distance distribution functions of the particles p(r). Molecular weights (MM) were estimated by comparison of the calculated forward scattering I(0) of the samples with that of a standard solution of bovine serum albumin (MM 66 kDa). The excluded volume of the hydrated protein molecule (V_p_) was computed by the Porod approximation [62]:(1)$$V_{p}=\frac{2\pi^{2}I\left(0\right)}{\int I_{\exp}\left(s\right)s^{2}ds}$$

Rigid-body modeling was performed using CORAL [48] in order to generate the approximate conformations of the missing pro-peptide regions, by keeping fixed the obtained high-resolution 3D model in solution of hBDNF [44] and by imposing P2 symmetry. Ten independent CORAL runs were performed for each data set. The resulting models were superimposed using the program SUPCOMB [63] and averaged using DAMAVER [64] to identify the most typical models representing the two protein variants in solution whose similarity was estimated by the normalized spatial discrepancy parameter (NSD) [63] obtained from DAMAVER. NSD values ≤ 1.0 are expected for similar models. The pro-peptide flexibility and size distribution of possible conformers were quantitatively assessed by the ensemble optimization method (EOM) [53]. A total of 10,000 randomized conformations were initially generated for hproBDNF WT and V66M, respectively, based on amino acid sequence and 3D-structural model of hBDNF used as a rigid body by imposing the native conformation option and applying P2 symmetry. The scattering profiles of these randomly generated conformations, calculated by CRYSOL [65], were compared and representative structures, whose calculated scattering curve fits the experimental measured scattering curve, were selected by a genetic algorithm. Multiple runs of EOM were performed to yield the R_g_ distributions in the selected ensembles. Agreement between the calculated scattering curve for the atomic rigid body models and the experimental scattering data was assessed via χ^2^ score.

## Figures and Tables

**Figure 1 ijms-23-06596-f001:**
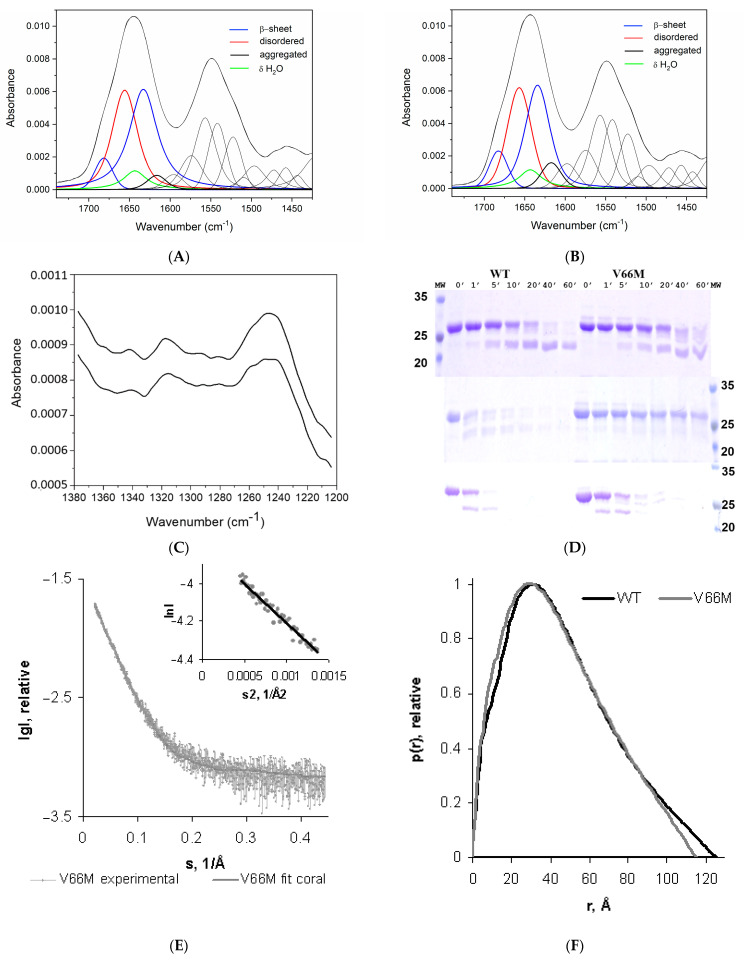
V66M polymorphism induces a conformational change in hproBDNF. Deconvolution of the FT-IR amide I and amide II bands of hproBDNF WT (**A**) and V66M variant (**B**), respectively. (**C**) Amide III region of WT (upper spectrum) and V66M protein (lower spectrum). (**D**) Coomassie Blue stained SDS-PAGE 12% gels of the progression of the digestion of hproBDNF WT and V66M variants incubated with thermolysin (upper panel), actinase (middle panel), and trypsin (lower panel). Undigested protein served as the zero-time point (0). (**E**) Experimental SAXS patterns of hproBDNF V66M. The plots display the logarithm of the scattering intensity as a function of momentum transfer (s). The corresponding Guinier plot is shown in the inserts. (**F**) Comparison of the derived distance distribution functions of hproBDNF WT (black trace) and V66M (grey trace).

**Figure 2 ijms-23-06596-f002:**
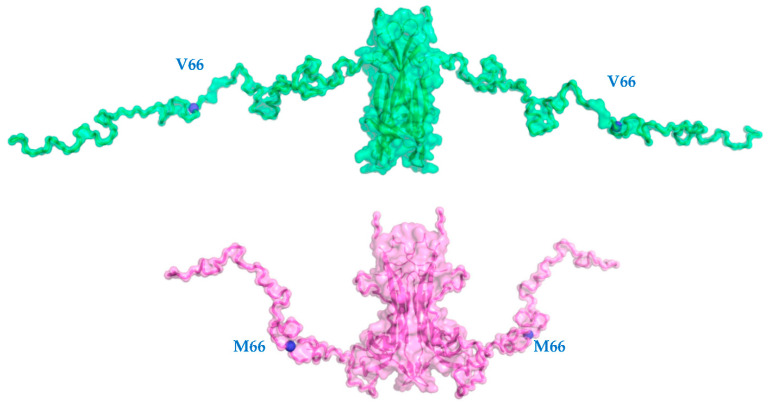
Typical rigid body models of hproBDNF WT (green) compared to the one reported for the V66M variant (violet) [44], obtained by CORAL (semitransparent surfaces) applying P2 symmetry; cartoon representations of the modeled structure of mature hBDNF, the positions of V66 and M66 are depicted by blue dots in the hproNGF WT, and V66M mutant, respectively. The two representations have been obtained by a 90° rotation around the y-axis. Figure produced by Pymol (Schrödinger LLC 2010).

**Figure 3 ijms-23-06596-f003:**
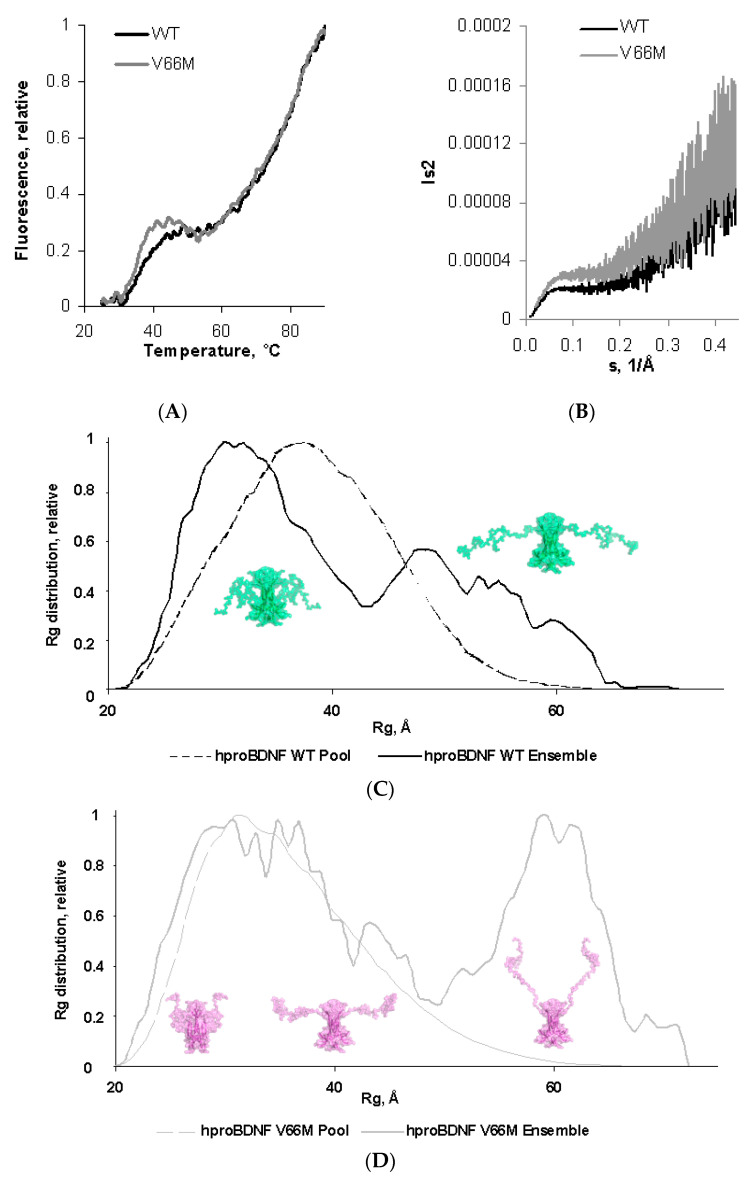
V66M polymorphism induces an increased plasticity in hproBDNF in solution. (**A**) Thermal denaturation assay of hproBDNF WT (black trace) and of the V66M variant (grey trace) by DSF with SYPRO© dye. Experiments were performed in triplicate: the mean curve is shown for clarity. (**B**) Kratky plots derived from SAXS measurements of hproBDNF WT and V66M variants; (**C**,**D**) R_g_ distributions obtained by EOM, imposing P2 symmetry for hproBDNF WT (black trace) and V66M (grey trace), respectively. The distributions for the initial random pools of models and the selected ensembles are shown by dots and solid lines, respectively. The representative conformations (semitransparent surfaces) are shown near the distributions: compact on the left, intermediate in the middle, and extended on the right. Folded protomers of mature hBDNF are depicted as cartoons. Figure produced by Pymol (Schrödinger LLC 2010).

**Table 1 ijms-23-06596-t001:** SAXS structural parameters: radius of gyration (R_g_), maximum dimension (D_max_), Porod volume (V_p_), and Molecular Mass (MM). D_max_ was obtained from the P(r) distribution using GNOM; I(0) (scattering intensity) was obtained from the scattering data by the Guinier analysis. MM was estimated by comparing the I(0) intensity of the known MM bovine serum albumin (BSA) standard.

Data Collection Parameters	hproBDNF V66M	hproBDNF
Instrument	P12 (PETRA III)	P12 (PETRA III)
Beam geometry (mm^2^)	0.2 × 0.12	0.2 × 0.12
Wavelength (Å)	1.24	1.24
*s* range (Å^−1^)	0.003–0.445	0.003–0.445
Concentration range (mg/mL)	0.14–1.45	0.10–1.48
Temperature (K)	283	283
**Structural parameters**		
I(0) (A.U.) [from p(r)]	0.02100 ± 0.0002	0.0177 ± 0.0002
R_g_ (Å) [from p(r)]	35.5 ± 0.05	37.2 ± 0.06
I(0) (A.U.) [from Guinier]	0.02263 ± 0.0002	0.0182 ± 0.0002
R_g_ (Å) [from Guinier]	35.5 ± 0.05	37.1 ± 0.06
D_max_ (Å)	115 ± 5	125 ± 6
V_p_ Porod volume estimate (Å^3^)	85,675 ± 6000	86,750 ± 6000
**Molecular mass determination (Da)**		
MM [from I(0)]	55,440 ± 6000	48,048 ± 6000
MM [from Porod volume]	50,400 ± 6000	51,030 ± 6000
Calculated monomeric MM from sequence	25,930	25,930
**Software employed**		
Primary data reduction	PRIMUS	PRIMUS
Data processing	GNOM	GNOM
Rigid body modeling	CORAL/EOM	CORAL/EOM
Validation and averaging	DAMAVER	DAMAVER
Three-dimensional graphic representations	PYMOL	PYMOL

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
