# Peer review of "Untangling the Conformational Plasticity of V66M Human proBDNF Polymorphism as a Modifier of Psychiatric Disorder Susceptibility"

_ijms, 2022, doi:10.3390/ijms23126596_

Round 1
Reviewer 1 Report
The study provided novel information on the differences in conformational flexibility, structural plasticity, and degradation of hproBDNF and the hproBDNF V66M variant. The findings would be of interest to researchers investigating the functional significance of the V66M human proBDNF polymorphism for susceptibility to mental disorders. I suggest stating explicitly the limitations of the study and describing directions for future research to confirm the functional significance of the detected differences.
Author Response
We sincerely express our deepest gratitude to the Reviewer for the positive responses and for the concise summary of our work and for the useful comment.
Appropriate amendments have been included in the revised manuscript.
Thus, as Reviewer #1 suggested, we have introduced at the end of the section Conclusions, the following sentences in order to state explicitly the limitations of the study and describing directions for future research to confirm the functional significance of the detected differences.
“Although we are aware of the functional in vivo limitations of the present investigation, as on the other hand for most of structural studies, these in vitro results strongly support and strengthen the functional differences and the associated behavioral consequences of the V66M hproBDNF polymorph.
Ongoing structural studies of a recently discovered and less characterized hproBDNF SNP, i.e., M122T, found in a small consanguineous family with intellectual disability [56], would allow us to confirm our present in vitro findings i.e. how a single amino acid mutation could lead to profound changes in structural stability and conformational flexibility. Furthermore, Förster Resonance Energy Transfer experiments should help elucidating differences in the dynamics of intra-molecular interactions of hproBDNF WT and of its pathological variant/s, in living cells.
We plan to complement our reported proteolytic sensitivity studies by exploiting physiological proteases involved in hproBDNF maturation, i.e., prohormone convertase, furin, plasmin and MMP-7.”
Reviewer 2 Report
This article entitled “Untangling the conformational plasticity of V66M human proBDNF polymorphism as a modifier of psychiatric disorder susceptibility” by Covaceuszack et al. reports series of structural analyses of human proBDNF and its V66M variant, known as risk factors for many neurological disorders. The authors revealed striking differences between the two variants that are consistent with the previously reported functional differences. From protein engineering point of view, this is an interesting example how a single amino acid mutation could lead to profound changes in structural stability. As the impact of this SNP to human brain functions is well-known to neuroscientists, this study would attract a wide range of readership. Only a few suggestions for clarification are listed below:
1. It would be helpful to indicate the location of the residue 66 in Fig. 2.
2. The difference in the lower transition point (Tm) shown in Fig. 3A should be emphasized, since this predicts a critical difference in the expected stability of two variants at body temperature. Related to this matter, it requires some explanation how to decide the temperature in Table 1.
3. In line 238, hydrogen bonds and salt bridges were listed as effectors of instability. However, losing the local hydrophobic interaction that Val residue had could also be important. Any reason to focus on H-bonds and salt bridges?
Author Response
We sincerely express our deepest gratitude to the Reviewer for the positive response and for the concise summary of our work and for the useful comment.
Appropriate amendments have been included in the revised manuscript.
In details:
- 1. It would be helpful to indicate the location of the residue 66 in Fig. 2.
Figure 2 has now been amended by highlighting the location of the residue 66 by a blue dot and appropriate labels.
- The difference in the lower transition point (Tm) shown in Fig. 3A should be emphasized, since this predicts a critical difference in the expected stability of two variants at body temperature
- In line 238, hydrogen bonds and salt bridges were listed as effectors of instability. However, losing the local hydrophobic interaction that Val residue had could also be important. Any reason to focus on H-bonds and salt bridges?
We thank the Reviewer for pointing out the likely relevance of hydrophobic effects.
Accordingly, we changed the sentence in:
“It is tempting to hypothesize that the structural plasticity of the hproBDNF V66M variant, inferred by physicochemical factors such as the re-arrangemet of the hydrogen bonding networking, as well as hydrophobic and/or salt bridges intra-molecular interactions, might in turn confer lower thermostability. Overall, these may be especially critical in vivo, being the Tm of hproBDNF V66M variant well below the human body temperature. “
Related to this matter, it requires some explanation how to decide the temperature in Table 1.
The biophysical characterization of hproBDNF and of its V66M variant pinpointed the prodomains to be IDRs. Therefore, SAXS experiments were carried out at 10°C (283 K), in order to avoid the complexity of mixed folded and unfolded prodomain populations likely occurring at 37°C, as highlighted by the thermal denaturation assay.